# A Pilot Study of Seamless Regeneration of Bone and Cartilage in Knee Joint Regeneration Using Honeycomb TCP

**DOI:** 10.3390/ma14237225

**Published:** 2021-11-26

**Authors:** Kiyofumi Takabatake, Hidetsugu Tsujigiwa, Aki Yoshida, Takayuki Furumatsu, Hotaka Kawai, May Wathone Oo, Keisuke Nakano, Hitoshi Nagatsuka

**Affiliations:** 1Department of Oral Pathology and Medicine Graduate School of Medicine, Dentistry and Pharmaceutical Science, Okayama University, Okayama 700-8525, Japan; gmd422094@s.okayama-u.ac.jp (K.T.); de18018@s.okayama-u.ac.jp (H.K.); p1qq7mbu@s.okayama-u.ac.jp (M.W.O.); keisuke1@okayama-u.ac.jp (K.N.); jin@okayama-u.ac.jp (H.N.); 2Department of Life Science, Faculty of Science, Okayama University of Science, Okayama 700-0005, Japan; 3Department of Orthopaedic Surgery, Okayama University Hospital, Okayama 700-8525, Japan; ayo@md.okayama-u.ac.jp (A.Y.); matino@md.okayama-u.ac.jp (T.F.)

**Keywords:** cartilage formation, bone formation, honeycomb TCP, knee joint regeneration, seamless regeneration

## Abstract

The knee joint is a continuous structure of bone and cartilage tissue, making it difficult to regenerate using artificial biomaterials. In a previous study, we succeeded in developing honeycomb tricalcium phosphate (TCP), which has through-and-through holes and is able to provide the optimum microenvironment for hard tissue regeneration. We demonstrated that TCP with 300 μm pore diameters (300TCP) induced vigorous bone formation, and that TCP with 75 μm pore diameters (75TCP) induced cartilage formation. In the present study, we regenerated a knee joint defect using honeycomb TCP. 75TCP and 300TCP were loaded with transforming growth factor (TGF)-β alone or bone morphogenic protein (BMP)-2+TGF-β with or without Matrigel and transplanted into knee joint defect model rabbits. 75TCP showed no bone or cartilage tissue formation in any of the groups with TGF-β alone and BMP-2+TGF-β with/without Matrigel. However, for 300TCP and BMP-2+TGF-β with or without Matrigel, vigorous bone tissue formation was observed in the TCP holes, and cartilage tissue formation in the TCP surface layer was continuous with the existing cartilage. The cartilage area in the TCP surface was larger in the group without Matrigel (with BMP-2+TGF-β) than in the group with Matrigel (with BMP-2+TGF-β). Therefore, honeycomb TCP can induce the seamless regeneration of bone and cartilage in a knee joint.

## 1. Introduction

In recent years, artificial biomaterials have been used for reconstruction after surgical resection of tumors and for bone tissue defects caused by trauma. A variety of artificial biomaterials having high biocompatibility such as hydroxyapatite (HA), calcium, and β-tricalcium phosphate (β-TCP) have already been clinically applied for bone tissue regeneration [1,2,3,4,5]. Bone tissue regeneration using artificial biomaterials has been reported in various studies and clinically applied, and these have been further developed and used clinically with new composite materials, such as hydroxyapatite and collagen composites [6,7,8].

For cartilage tissue regeneration, Brittberg et al. developed an autologous cultured chondrocyte transplantation method [9], and recently cartilage tissue regeneration by cell transplantation using bone marrow mesenchymal stem cells and synovial cells, as well as cartilage regeneration research using iPS cells, has progressed [10,11,12,13,14]. In cartilage tissue engineering, synthetic biodegradable polymers such as polylactic acid, polylactic acid-glycolic acid copolymers, and collagen sponges have been used as scaffolds, and gels with a chemical composition similar to that of cartilage matrix, such as type II collagen plus hyaluronic acid, have been developed as scaffolds for cartilage culture [15]. Because chondrocytes are resistant to environments with low nutrients and oxygen, they can grow well in 3D scaffold materials and produce extracellular matrix. In vitro, cartilage tissue regeneration is also easy because cartilage tissue does not require the introduction of blood vessels and does not require an environment in which multiple cell types work together like internal organs.

However, the actual cartilage of the knee joint has a special structure in which bone tissue and cartilage tissue are continuous, and in regeneration with artificial biomaterials, the joint of articular cartilage tissue with subchondral bone is problematic. Furthermore, in contrast to bone tissue, which is rich in blood vessels and has a strong regenerative capacity, articular cartilage is a tissue with very poor regenerative capacity [16]. Damage to the cartilage on the joint surface can easily lead to osteoarthritis, and it is extremely difficult to regenerate the joint surface with vitreous cartilage again. For knee joints, metal prostheses have been clinically applied due to their durability; however, the risks of infection, metal allergy, and other problems have not yet been solved [17].

For tissue regeneration using biomaterials, three key factors are essential: cells, extracellular matrix (ECM), and growth factors. In recent years, the composition, physical and chemical signals, pore size, and mechanical properties of gradient scaffolds have especially attracted interest [18,19]. Among these, we have focused on the importance of the pore size in the induction of hard tissue cell differentiation and have been developing a novel biomaterial, honeycomb TCP, which has a honeycomb-like arrangement of linear through holes. As a result, we have proved that honeycomb TCP has the ability to induce bone tissue by implanting it in the head region of rats [20,21,22,23]. These studies showed that vigorous bone tissue formation occurred in honeycomb TCP containing through-and-through holes with diameters of 300 μm in zygomatic and skull defect model rats, suggesting its clinical applicability. In addition, by changing the pore diameter of honeycomb TCP, we have reproduced the hard tissue formation microenvironment and succeeded in specifically inducing and forming cartilage. Specifically, cartilage formation was observed in honeycomb TCP with a 75-μm pore size in rat femoral muscle [24]. Therefore, this honeycomb β-TCP, which can reproduce the microenvironment, can control cell differentiation by changing the geometrical structure and enabling differentiation of chondrocytes and osteoblasts.

The purpose of this study was to investigate whether it is possible to regenerate cartilage and bone tissue in a seamless manner by providing an appropriate microenvironment with honeycomb TCP for knee joint cartilage defects, which have a special structure with continuous cartilage and bone tissue.

## 2. Materials and Methods

### 2.1. Experimental Animals and Ethics

NZW (New Zealand White) rabbits (SHIMIZU Laboratory Supplies, Co., Ltd., Kyoto, Japan) were used in this study. The Animal Experiment Control Committee of Okayama University approved this study (OKU-2020523).

### 2.2. Preparation of Honeycomb TCP Scaffolds

Honeycomb TCP used in ectopic experiments was pressed in a cylindrical mold containing through-and-through holes with diameters of 75 μm (75TCP) and 300 μm (300TCP) (Figure 1).

This TCP was calcinated by heating to 1200 °C. Details of TCP manufacturing have been described previously [23]. Each TCP structure was sterilized by autoclaving and loaded with transforming growth factor (TGF)-β alone or bone morphogenic protein (BMP)-2+TGF-β. Honeycomb TCPs were loaded with BMP-2 diluted to a final concentration of 125 ng with or without Matrigel^®^ (BD Biosciences, Inc., Franklin Lakes, NJ, USA) and loaded with TGF-β diluted to a final concentration of 1000 ng with or without Matrigel^®^ (BD Bioscience). 

### 2.3. Implantation and Histological Examination

Experimental animals were anesthetized with 50 mg/kg ketamine intramuscularly and 2% isoflurane inhalation. Then, bilateral distal femoral skin incisions of 5 cm were made to expose the femoral articular cartilage, and two holes of 3 mm in diameter and 5 mm in depth were made perpendicular to the cartilage tissue at the lateral and medial sides of the articular cartilage using a dental engine. The honeycomb TCPs were implanted into the formed cartilage holes and the wounds were closed.

At four weeks, the animals were euthanized with an overdose of carbon dioxide and were removed. All samples were fixed by 4% paraformaldehyde and decalcified with 10% ethylenediaminetetraacetic acid (EDTA). After decalcification, the samples were embedded in paraffin, sectioned at 5 μm in thickness, and stained by hematoxylin-eosin (HE stain) and Safranin O following standard histological protocols.

### 2.4. Cartilage Tissue Formation Evaluation by Area Measurement

To quantify the cartilage tissue formation area, cartilage tissue formation area on the surface layer of 300TCPs were measured in each sample in HE-stained specimens (200× magnification, *n* = 3) using Image J software (NIH, Bethesda, Rockville, MD, USA).

### 2.5. Statistical Analysis

All statistical analyses were conducted using GraphPad Prism 9 (GraphPad Software, Inc.). Data are presented as the mean ± standard deviation (SD). One-way ANOVA was used to compare two variables with Tukey’s post hoc test. *p* < 0.05 was considered to indicate a statistically significant difference.

## 3. Results

### 3.1. Histological Findings for 75TCP

For 75TCP, Matrigel remained in the TCP pores, and no bone tissue formation was observed in the pores with Matrigel only (Figure 2A,B), BMP-2 with Matrigel (Figure 2C,D), and TGF-β+BMP-2 with Matrigel (Figure 2E,F). A small amount of cartilage tissue formation was observed at the knee joint side of TCP for Matrigel only and for TGF-β+BMP-2 with Matrigel groups. Cartilage formation was not continuous with existing cartilage. 

### 3.2. Histological Findings for 300TCP

For 300TCP added to TGF-β without Matrigel, there was no inflammatory cell infiltration around the TCP, and hard tissue formation was continuous with the knee joint defect, indicating high biocompatibility (Figure 3A). At the site where the TCP contacted with the existing bone marrow, bone formation was observed to be continuous from the existing bone. In the central part of the TCP pores, bone was formed to add to the TCP wall, and on the cartilage side of TCP, fibrous connective tissue formation was observed in the TCP holes (Figure 3B). In the superficial layer of the TCP, fibrous connective tissue formation was present, and the formation of thin cartilage tissue was observed in the fibrous connective tissue formed in the TCP surface layer (Figure 3C,D). 

For 300TCP with TGF-β and Matrigel, the superficial layer of TCP was covered with fibrous connective tissue, and cartilage tissue formation was observed in some areas. Although cartilage tissue regeneration was observed continuously from the existing surrounding cartilage tissue, no cartilage tissue was observed in the central part of the cartilage tissue defect, and fibrous connective tissue was observed. Residual Matrigel was observed in some fibrous connective tissue (Figure 4C,D).

Osteogenesis was similar to the group without Matrigel, and bone formation from existing bone was observed in the area where the TCP was in contact with existing bone marrow (Figure 4B). On the cartilage side of TCP, the formation of fibrous connective tissue was observed in the TCP holes, and areas where no cell component invasion occurred were observed (Figure 4A). 

For 300TCP with TGF-β+BMP without Matrigel, there was vigorous bone tissue formation in the TCP holes and vigorous cartilage tissue formation on the cartilage side of the TCP holes (Figure 5A).

There was extensive regenerated cartilage tissue from the existing cartilage tissue to the center of the TCP, and there was bone tissue formation between the TCP and cartilage. The fibrous connective tissue in the upper part showed a tendency to differentiate into cartilage based on its staining properties and the formed cartilage tissue was stained with Safranin O (Figure 5C,D).

Osteogenesis was observed in continuity with the existing bone marrow, and bone formation with a bone marrow-like structure similar to that in a living body was observed up to the top of the TCP (Figure 5A,B).

In 300TCP with TGF+BMP and Matrigel, extensive chondrogenic differentiated tissue was formed in the superficial layers of the TCP. In the small part of cartilage tissue formation, residual Matrigel was also observed. However, no bone tissue formation was observed in the TCP pores near the TCP surface layer (Figure 6C,D).

Similar to the group without Matrigel, bone formation was observed in continuity with the existing bone marrow, and bone formation with a bone marrow-like structure similar to that in a living body was observed up to the top of the TCP (Figure 6A,B).

### 3.3. Quantitative Examination of Cartilage Tissue Formation in 300TCP

We quantitatively examined the area of cartilage tissue formation area on the surface layer of 300TCPs. In the 300TCP with TGF-β only, the area of cartilage tissue formation tended to be larger in the group with Matrigel than in the group without Matrigel, however there was no significant difference. On the other hand, in 300TCP with TGF-β+BMP-2, cartilage tissue formation area on the surface of 300TCP was significantly larger in the group without Matrigel than in the group with Matrigel. The cartilage tissue formation area was larger in 300TCP added with TGF-β+BMP-2 group than in 300TCP added with TGF alone group with or without the addition of Matrigel (Figure 7).

## 4. Discussion

Tissue architecture involving more than one cell type is a major challenge in tissue engineering [25]. In osteochondral tissue engineering, an ideal scaffold should be a biomimetic of ECM and address the requirements of different tissues [26]. In this study, a honeycomb TCP was used to create a suitable microenvironment for continuous bone and cartilage tissue regeneration in the knee joint.

In this study, 300TCP exhibited a good connection between bone formation and cartilage formation in the knee joint. Bone tissue formation was also observed from the tibial side (deep side) of the TCP, and cartilage tissue and fibrous connective tissue were formed in the TCP surface layer (knee joint side), suggesting that cells were supplied from the bone marrow direction. In osteoarthritis of the knee, mesenchymal stem cells are mobilized from the synovium into the joint fluid and adhere to the degenerated cartilage to promote the production of cartilage matrix. It has been reported that synovium-derived mesenchymal stem cells are useful as a cell source for cartilage regeneration because of their high chondrogenic differentiation potential and reliable cell number [27]. Thus, in many studies, synovium contains undifferentiated mesenchymal stem cells, which are expected to be applied to regenerative medicine [28,29]. However, in the case of simultaneous repair of bone and cartilage tissues using artificial biomaterials, as in this study, the cell supply route from the bone marrow is considered to be more important than the synovium, and the geometric structure of the artificial biomaterials suitable for cell migration and aggregation is considered to be important. In our previous study, we demonstrated that a geometric structure in which the direction of the TCP pore is horizontal from the cell source is advantageous [20,21]. The present study also suggests that parallelism between the long axis of the tibial bone marrow and the through hole of the TCP may be important. The present study also showed that a TCP pore size of 75 μm was unfavorable for cell infiltration from the bone marrow side, and no bone and cartilage tissue formation was observed.

In the 300TCP surface layer, cartilage tissue formation was observed in continuity with the existing cartilage tissue. In general, hard tissue formed by artificial biomaterials undergoes the process of endochondral ossification, which induces the differentiation of cartilage tissue into bone tissue. However, the results of this study suggest that the hard tissue formed on the surface layer of TCP remains cartilage, because the chondrocytes of existing cartilage tissue synthesize the vascular invasion inhibitory factor chondromodulin-1 (ChM-1) [30,31]. ChM-I, an ECM protein unique to cartilage that inhibits angiogenesis, is thought to play the most important role in cartilage tissue, and ChM-I was initially discovered as a growth factor of chondrocytes.

In this study, TGF alone, which is a chondroinductive factor, did not induce bone tissue in the TCP or cartilage tissue on the TCP surface layer. However, in the BMP+TGF group, bone tissue was vigorously induced in the TCP and cartilage tissue was induced on the TCP surface layer. In osteochondral tissue engineering, growth factors can promote osteochondral tissue regeneration [32,33]. TGF-β1 can stimulate mesenchymal stem cell (MSC) proliferation and induce ECM production [34], and BMP-2 can stimulate chondrogenesis and osteogenesis differentiation of MSCs [35]. The spatially controlled and localized delivery of multiple growth factors from TCP could direct the differentiation of bone marrow-derived MSCs to obtain a complex tissue. This is because both chondrocytes and osteoblasts differentiate from MSCs, and the first stage of differentiation is the aggregation of MSCs. At this stage, the transcription factor Sox-9 acts, and Sox-5 and Sox-6 are also necessary for this stage. In the process of chondrocytes becoming hypertrophic chondrocytes, Runx2 functions, and BMPs and TGF-β play an important role as secreted factors that promote these processes [36,37,38].

Various materials are currently used as carriers for local delivery of BMP-2 or TGF-β for in vivo bone formation. When growth factor alone is impregnated into an artificial biomaterial, growth factor rapidly diffuses away from the implantation area, so a delivery carrier is required. Matrigel is widely used as a carrier for growth factors in bone tissue regeneration experiments, and Matrigel has BMP-2 retention and has been used in many osteoinductive animal experiments. However, in the present study, the group with Matrigel had worse hard tissue formation than the group without Matrigel. Matrigel remained in the groups where it was added even after four weeks, suggesting that the remaining Matrigel might be an obstacle to cell invasion.

## 5. Conclusions

By using a novel biomaterial honeycomb TCP, we succeeded in inducing bone tissue in the bone equivalent area and cartilage tissue in the cartilage equivalent area of the knee joint. Our study indicates that honeycomb TCP is an excellent artificial biomaterial that can serve in knee joint regeneration.

## Figures and Tables

**Figure 1 materials-14-07225-f001:**
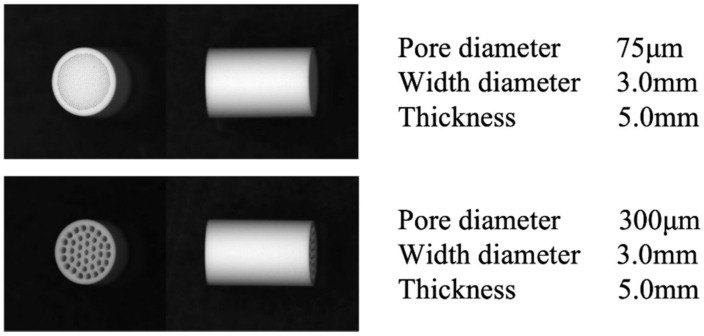
Honeycomb TCP structures used in the experiments.

**Figure 2 materials-14-07225-f002:**
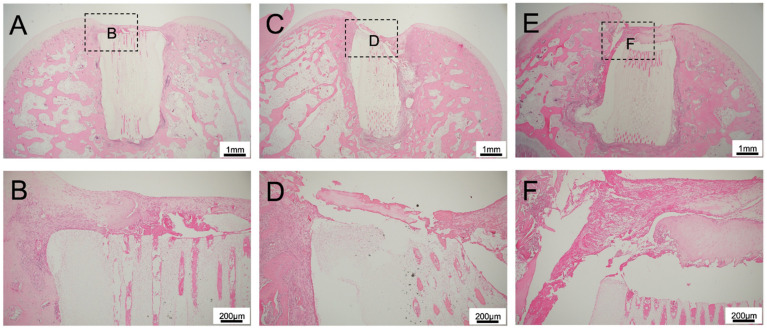
Histological findings of 75TCP experiments. (**A**) Histological finding of the Matrigel-only group at low power magnification, and (**B**) Histological finding of the Matrigel-only group at high power magnification. (**C**) Histological finding of the BMP-2 with Matrigel group at low power magnification, and (**D**) Histological finding of the BMP-2 with Matrigel group at high power magnification. (**E**) Histological finding of the TGF-β+BMP-2 with Matrigel group at low power magnification, and (**F**) Histological finding of the TGF-β+BMP-2 with Matrigel group at high power magnification. In all experimental groups, Matrigel remained in the TCP pores, and no bone tissue formation was observed in the pores.

**Figure 3 materials-14-07225-f003:**
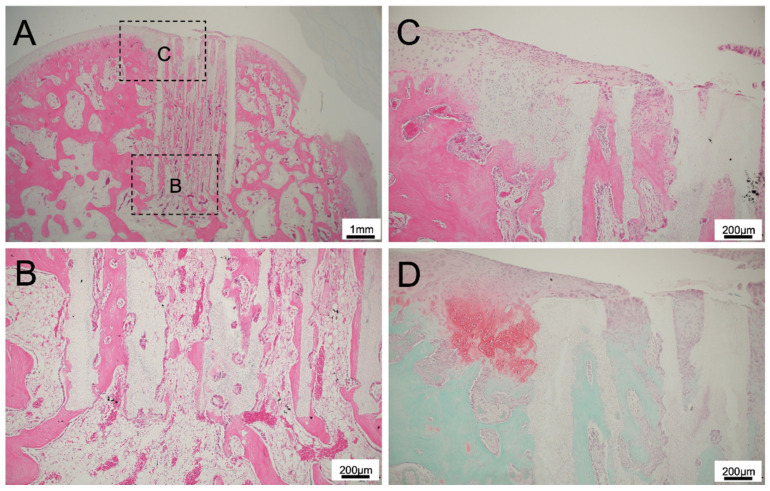
(**A**) Histological finding of 300TCP with TGF-β without Matrigel at low power magnification. (**B**,**C**) Histological finding of 300TCP with TGF-β without Matrigel at high power magnification. At the site where the TCP contacted with the existing bone marrow, bone formation was observed to be continuous from the existing bone. In the superficial layer of the TCP, fibrous connective tissue formation was present, and the formation of thin cartilage tissue was observed in the fibrous connective tissue formed in the TCP surface layer. (**D**) Safranin O staining showed slight cartilage formation in the superficial layer of the TCP.

**Figure 4 materials-14-07225-f004:**
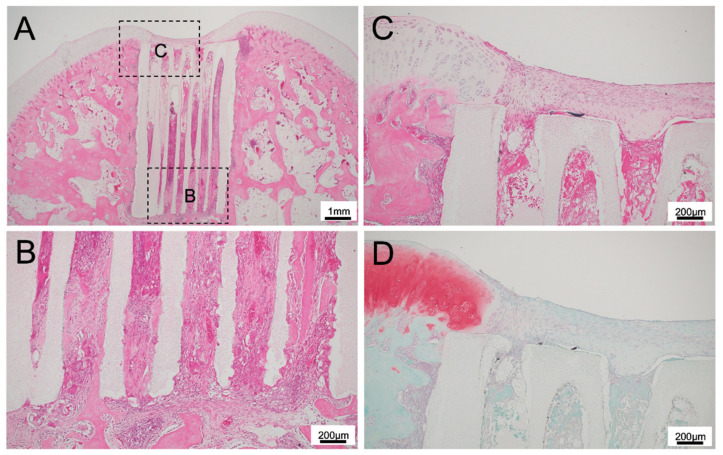
(**A**) Histological finding of 300TCP with TGF-β with Matrigel at low power magnification. (**B**,**C**) Histological finding of 300TCP with TGF-β with Matrigel at high power magnification. Bone formation from existing bone was observed in the area where the TCP was in contact with existing bone marrow. The superficial layer of TCP was covered with fibrous connective tissue, and cartilage tissue formation was observed in some areas. (**D**) Safranin O staining showed slight cartilage formation in the superficial layer of the TCP.

**Figure 5 materials-14-07225-f005:**
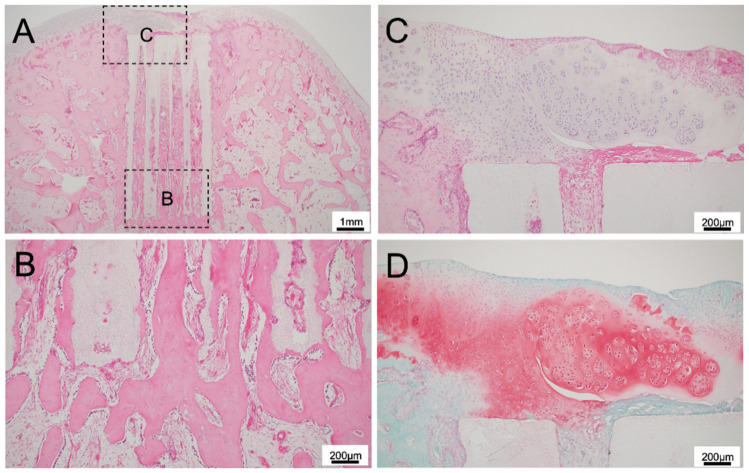
(**A**) Histological finding of 300TCP with TGF-β+BMP without Matrigel at low power magnification. (**B**–**D**) Histological finding of 300TCP with TGF-β+BMP without Matrigel at high power magnification. Bone formation from existing bone was observed in the area where the TCP was in contact with existing bone marrow. Cartilage regeneration was also observed, and spherical cartilage stained by Safranin O was also observed in the superficial layer of the TCP.

**Figure 6 materials-14-07225-f006:**
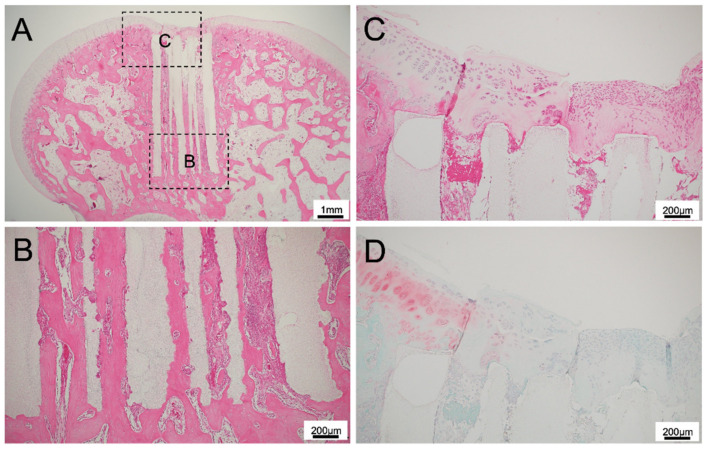
(**A**) Histological finding of 300TCP with TGF-β+BMP-2 with Matrigel at low power magnification. (**B**–**D**) Histological finding of 300TCP with TGF-β+BMP-2 with Matrigel at high power magnification. Bone formation from existing bone was observed in the area where the TCP was in contact with existing bone marrow. Cartilage regeneration as stained by Safranin O was also observed in the superficial layer of the TCP.

**Figure 7 materials-14-07225-f007:**
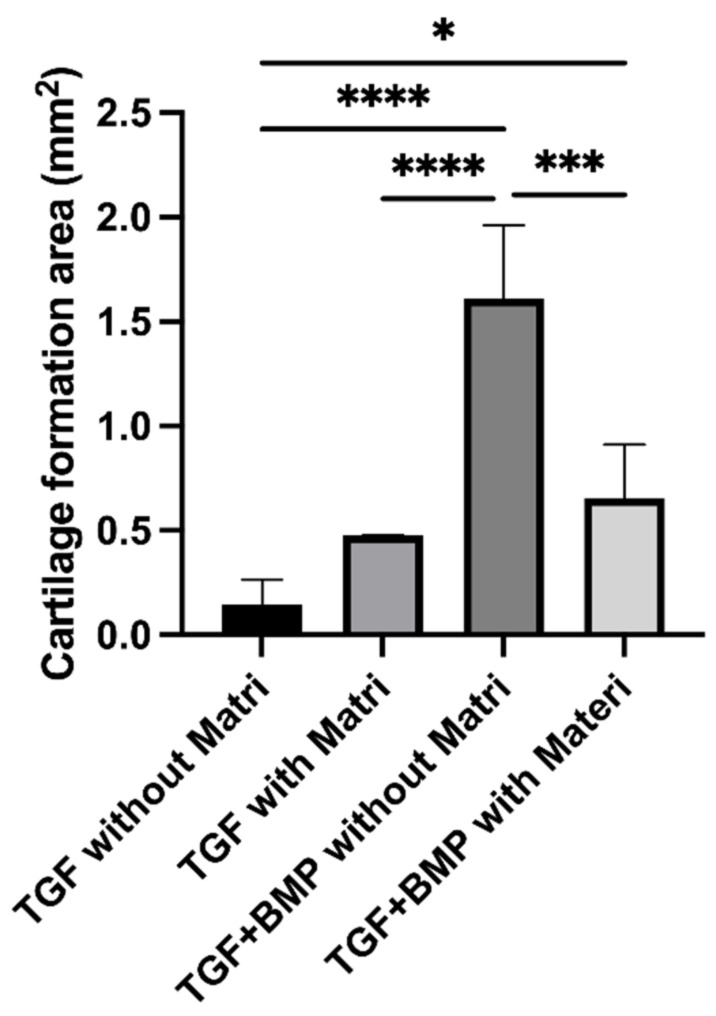
Quantitative analysis of cartilage formation area on the surface of 300TCP. Data are presented as the mean ± standard deviation (SD). One-way ANOVA was used to compare two variables with Tukey’s post hoc test. *: *p* < 0.05, ***: *p* < 0.0001, ****: *p* < 0.00001.

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
