# Peer review of "A Pilot Study of Seamless Regeneration of Bone and Cartilage in Knee Joint Regeneration Using Honeycomb TCP"

_materials, 2021, doi:10.3390/ma14237225_

Round 1

Reviewer 1 Report

  • Please notify and determine with the statistical test if there is any relationship between groups.

  • Did the release of TGF β or BMP calculate? In our opinion, it should be considered and compare the releasing of growth factors used in the study according to different pore sizes designed.

  • The authors used matrigel as a carrier for growth factors. Please mention the time witch these factors could remain in the matrix after transplantation.

Author Response

Please notify and determine with the statistical test if there is any relationship between groups.

→We have analyzed the cartilage tissue formation evaluation by measurement with the statistical test in Fig. 7.

Did the release of TGF β or BMP calculate? In our opinion, it should be considered and compare the releasing of growth factors used in the study according to different pore sizes designed.

→Thank you for your suggestion. We have never calculated the release of TGF-β or BMP-2. However, in our previous studies, we have investigated bone and cartilage tissue formation using various pore sizes of TCP with various concentrations BMP-2. We believe that the amount of hard tissue formation due to pore size and the amount of growth factor impregnated is more important than the amount of growth factor released.

The authors used matrigel as a carrier for growth factors. Please mention the time witch these factors could remain in the matrix after transplantation.

→We have added to the results of histological findings of residual Matrigel. In this experiment, it is not known how long the growth factors remain in the Matrigel. In this experiment, in the TGF+BMP group, the area of cartilage tissue formation was larger in the group without Matrigel than in the group with Matrigel. This result suggests that Matrigel is necessary to retain growth factors, but that Matrigel may interfere with cell invasion. This content has already included in the end of Discussion section.

Reviewer 2 Report

The article represents an experimental study on bone and cartilage formation in a novel TCP based biomaterials with interconnected bone. There are several issues with the submission that need to be resolved (below) before the journal proceeds to considering the work for publication. There are also some minor issues with typos, clarity, grammatical spelling, vocabulary, not appropriately using/misuse of articles and prepositions, proper use of singular and plural forms, non-consistent tense and etc.

Introduction:

  • Please explain why others significance associated with use of metallic biomaterials, i.e., stress-shielding is the significant negative effect.
  • Please explain how the bigger pore size would affect the mechanical stability of biomaterials, especially for the ones in load-bearing applications.
  • According to the literature any biomaterials which is biocompatible and mechanical stable can enable bone-ingrowth to some extent, please elaborate what is the shortcoming of the current scaffold.

Methodology:

  • This study lacks proper statistical analysis for any meaningful conclusion. The authors should consider performing proper statistical test for all the comparisons they reported.
  • The authors should include/indicate the control group was used for this study and their justification for this selection.

Results/Discussion:

  • I was not able to comment on this section since the proper statistical analysis were missed.

Author Response

The article represents an experimental study on bone and cartilage formation in a novel TCP based biomaterials with interconnected bone. There are several issues with the submission that need to be resolved (below) before the journal proceeds to considering the work for publication. There are also some minor issues with typos, clarity, grammatical spelling, vocabulary, not appropriately using/misuse of articles and prepositions, proper use of singular and plural forms, non-consistent tense and etc.

Introduction:

Please explain why others significance associated with use of metallic biomaterials, i.e., stress-shielding is the significant negative effect.

→We have mention that prostheses have been clinically applied due to their durability; however, the risks of infection, metal allergy, and other problems have not yet been solved. We have modified the text.

Please explain how the bigger pore size would affect the mechanical stability of biomaterials, especially for the ones in load-bearing applications.

→Artificial biomaterials with porous structures generally have low physical properties, and in our TCP, the physical properties are expected to decrease as the pore size increases. However, in our previous study, honeycomb TCP was successfully used to reconstruct a zygomatic defect, an area that is constantly subjected to masticatory forces. Therefore, we believe that honeycomb TCP shows stable physical properties in load-bearing applications.

According to the literature any biomaterials which is biocompatible and mechanical stable can enable bone-ingrowth to some extent, please elaborate what is the shortcoming of the current scaffold.

→Although artificial biomaterials including honeycomb TCP have biocompatibility, their disadvantage is that they cannot be clinically applied at present in extensive bone defects due to their low physical properties, unlike metallic materials. It is necessary. for biomaterials to improve the physical properties in the future.

Methodology:

This study lacks proper statistical analysis for any meaningful conclusion. The authors should consider performing proper statistical test for all the comparisons they reported.

→We have analyzed the cartilage tissue formation evaluation by measurement with the statistical test in Fig. 7.

The authors should include/indicate the control group was used for this study and their justification for this selection.

→Thank you for your suggestion; however, previous studies have examined the tissue formation of bone and cartilage using various concentrations of BMP-2. As a result, no hard tissue formation was observed at all when Matrigel alone was used and no growth factors were impregnated, so we did not have a negative control without growth factors in this study.

Results/Discussion:

I was not able to comment on this section since the proper statistical analysis were missed.

→We have analyzed the cartilage tissue formation evaluation by measurement with the statistical test in Fig. 7.

Round 2

Reviewer 2 Report

The Statistical comparison should include a meaningful statistical test (i.e., T-test, ANOVA or etc) based on data distribution and sample size. All the significant p-values must be reported and statistical power analysis should be included. 

Author Response

The Statistical comparison should include a meaningful statistical test (i.e., T-test, ANOVA or etc) based on data distribution and sample size. All the significant p-values must be reported and statistical power analysis should be included. 

→We have added to the texts about the statistical analysis in Materials and Methods, and we have modified the figure legend of Fig. 7.